# Topic-XICL: Demonstration Selection with Topic Inference for Cross-lingual In-context Learning

## Abstract

Cross-lingual in-context learning (XICL) shows promise for adapting large language models (LLMs) to low-resource languages. Previous methods typically rely on off-the-shelf similarity-based approaches or task-specific retrievers trained with LLM feedback for demonstration selection. However, these methods often overlook important factors beyond a single criterion or can be resource-intensive. To address these challenges, we propose a novel approach called Topic-XICL, which leverages a latent topic model for demonstration selection. We assume that latent topic variables encapsulate information that more accurately characterizes demonstrations. By training this topic model on rich-resource language data with a compact LLM, we obtain more relevant demonstrations through topic inference and apply them for in-context learning across various LLMs. We evaluated our method on three multilingual tasks (XNLI, XCOPA, and TyDiQA-GoldP) using three models with 7 to 8 billion parameters (BLOOM, Qwen1.5, and Llama3.1). Our approach outperformed the baselines—random selection, semantic similarity, and clustering-based methods—on TyDiQA-GoldP, XCOPA, and XNLI by 3.32%, 2.47%, and 1.77%, respectively, while requiring only moderate additional resources.

## 1 Introduction

Large Language Models (LLMs) have demonstrated remarkable capabilities in natural language understanding across a wide range of NLP tasks in English (Lai et al., 2023; Bang et al., 2023; Zhang et al., 2023). Recent advancements have extended their multilingual functionalities (Shi et al., 2023; Cahyawijaya et al., 2023; Chen et al., 2023; Yang et al., 2024); however, achieving robust performance across multiple languages often requires substantial amounts of data for training or fine-tuning. In-context learning (ICL) (Brown et al., 2020; Scao et al., 2022; Lin et al., 2022) has emerged as a promising approach to improve the performance of LLM in low-resource languages.

The impressive comprehension abilities of LLMs in English have fueled interest in Cross-lingual In-Context Learning (XICL)(Winata et al., 2021; Lin et al., 2022; Asai et al., 2023; Cahyawijaya et al., 2024; Zhang et al., 2024), which leverages demonstrations from high-resource languages to guide learning in low-resource languages. However, the effectiveness of XICL relies heavily on the selection of demonstration examples (Zhao et al., 2021; Perez et al., 2021; Qin et al., 2023; Cahyawijaya et al., 2024). Researchers have proposed two primary approaches for demonstration selection: using off-the-shelf retrievers(Nie et al., 2023; Chang & Fosler-Lussier, 2023; Winata et al., 2023; Li et al., 2023; Cahyawijaya et al., 2024), such as BM25 or Sentence-BERT (Reimers & Gurevych, 2019), and training task-specific retrievers (Shi et al., 2022) using specially designed task signals, like feedback from LLMs. While task-specific retrievers may produce better results for certain LLMs, they often require access to model parameters or detailed output distributions, which can be costly to obtain and are typically unavailable for black-box LLMs (Sun et al., 2022). In contrast, off-the-shelf methods offer a lightweight solution by exploiting semantic similarity between input-label pairs, but they tend to overlook task-specific information and diversity.

As noted in Qin et al. (2023), the choice between similarity and diversity in demonstrations varies by task: diversity works better for tasks like commonsense reasoning and question answering, while

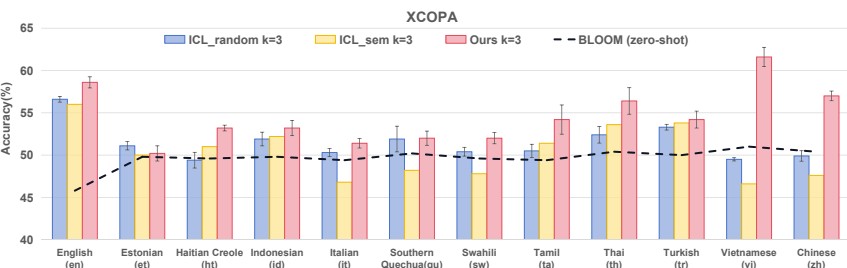

Figure 1: Accuracy scores for 12 languages from the XCOPA dataset (Gordon et al., 2012) using zero-shot inference (dashed line) and 3-shot in-context learning (ICL) with the BLOOM model (Scao et al., 2022) (7.1 billion parameters). $k$ denotes the number of demonstrations. "sem" refers to semantic-based selection, while "random" refers to random selection.

similarity is more effective for text classification. Fig. 1 highlights the challenge of balancing these two dimensions across languages. Semantically similar examples yield better results for Haitian Creole (ht), Tamil (ta), and Thai (th), whereas randomly selected diverse examples perform better for Italian (it), Quechua (qu), and Swahili (sw). When selecting demonstrations across languages, it's important to account for not just semantic similarity, but also factors like syntactic structure, task structure, and domain information. We collectively abstract these flexible factors as latent topic information, which encapsulates characteristics that better represent demonstrations, improving cross-lingual in-context learning.

Xie et al. (2022) examined in-context learning from a Bayesian inference perspective, while Wang et al. (2023) treated LLMs as topic models, applying this theory effectively in demonstration selection for classification tasks. Building on this, we extend Wang et al. (2023)'s approach to cross-lingual in-context learning across various tasks, proposing a demonstration selection algorithm based on topic inference (Topic-XICL), as shown in Fig. 2. Our method consists of two phases: **latent topic learning** and **demonstration selection**. In the latent topic learning phase, demonstration candidates from a rich-resource language are clustered into topics using the K-means algorithm, based on multilingual representations, and a topic model is trained using an LLM to capture nuanced topic information. Specifically, the candidate data for a task are grouped into $n$ topics, and for each topic, we introduce $c$ new tokens to extend the LLM's vocabulary. These tokens are concatenated with the input, allowing the LLM to update token embeddings and improve predictions. In the demonstration selection phase, topic inference is performed on the candidate data to select the $k$ most representative examples for each topic. For each target language input, its topic is determined by calculating semantic similarity with the candidate data, using the corresponding representative examples of its topic as context.

We trained the latent topic model on BLOOMZ-1b7 (Muennighoff et al., 2023) (with 1.7 billion parameters) and conducted cross-lingual ICL on two multilingual sentence-level tasks and one cross-lingual reading comprehension task. Our contributions are summarized as follows:

- We propose a cross-lingual demonstration selection algorithm based on topic inference (Topic-XICL), extending Bayesian inference theory to practical applications in cross-lingual ICL.

- Intuitively, the Bayesian theorem is primarily suited for classification tasks. To our knowledge, we are the first to apply it to non-classification tasks on XICL, and we have experimentally validated its effectiveness.

- Our topic-based demonstration selection method outperforms the strongest baselines across three models—BLOOM, Qwen1.5, and Llama3.1—by 3.32%, 2.47%, and 1.77% on average for TyDiQA-GoldP, XCOPA, and XNLI, respectively.

## 2 RELATED WORK

**Cross-lingual In-context learning** The cross-lingual nature of multilingual language models further enables the possibility of learning from a different language in-context without parameter updates, as demonstrated by the XICL method (Winata et al., 2021; Lin et al., 2022). Winata et al.

(2021) first showed that, given a few English examples as context, multilingual pre-trained language models (such as GPT (Radford et al., 2019) and T5 (Raffel et al., 2020)) can predict not only English test samples but also non-English ones. Lin et al. (2022) also found that their XGLM demonstrates strong cross-lingual capability, where using English prompts together with non-English examples yields competitive zero- and few-shot learning performance. Cahyawijaya et al. (2024) extensively studied XICL on some low-resource languages from four aspects: cross-lingual alignment, alignment formatting, label configuration, and cross-lingual retrieval, highlighting the importance of advancing ICL research. Our research mainly focuses on the aspect of cross-lingual retrieval to select demonstrations for XICL.

**Cross-lingual Demonstration Selection**  Different rich-resource language demonstrations yield varying XICL outcomes for target languages. Current cross-lingual retrieval methods fall into two categories: using off-the-shelf multilingual representations and leveraging LLM feedback signals. For example, Nie et al. (2023) conducts cross-lingual retrieval from labeled or unlabeled high-resource languages based on the semantic similarity of multilingual embeddings. Li et al. (2023) extended this to focus on zero-shot settings, revealing limitations for complex generation tasks. Tanwar et al. (2023) augmented prompts with cross-lingual semantic similarity demonstrations and in-context label alignment, but Cahyawijaya et al. (2024) identified shortcomings and introduced translation pairs for alignment. Additionally, Winata et al. (2023) emphasized semantic similarity by selecting the nearest examples from various sub-datasets for classification tasks. In contrast, Shi et al. (2022) proposed a retrieve-rerank framework for cross-lingual Text-to-SQL, using a bi-encoder to identify relevant exemplars, and then training a retriever by distilling the LLM's scoring function.

Training retrievers on specific task data and LLMs can be advantageous, but managing inaccessible parameters of black-box models is challenging. Our method trains using only accessible LLMs. Semantic similarity alone may not suffice for complex tasks, so we expect to integrate richer information into "latent topics," such as article types in question-answering tasks, question types, and the structural relationship between answers and articles. We use LLMs to mine this latent topic information and select demonstrations to enhance cross-lingual in-context learning.

**In-Context Learning with Bayesian inference**  Xie et al. (2022) provided a latent topic interpretation to explain in-context learning, showing that the in-context learning predictor approaches the Bayes optimal predictor as the number of demonstrations increases, assuming both pre-training and task-specific data follow Hidden Markov Models (HMM). However, the Markovian assumption about data generation limits empirical validation to synthetic data and toy models, raising questions about its applicability to natural language.

To bridge the gap between theoretical understanding and real-world LLM algorithms, Wang et al. (2023) developed a practical demonstration selection algorithm. Our method extends Wang et al. (2023) to an XICL setting. Unlike their approach, which treats each classification data as a topic, we perform semantic clustering on each task's data to obtain topics, making our approach applicable to a wider range of tasks. To our knowledge, this is the first attempt to use Bayesian theory for demonstration selection beyond classification.

## 3 METHOD

Based on the theoretical understanding and practical algorithm of Bayesian inference in ICL, we proposed a cross-lingual demonstration selection framework (as shown in Fig. 2) with topic inference to improve the performance of cross-lingual ICL for various tasks, not only classification tasks. First, we introduce the notations of problem setting and theoretical analysis of the problem. Then we describe the pipeline to learn latent topic embedding in Section 3.2 and the algorithm of demonstration selection in Section 3.3.

### 3.1 NOTATIONS AND PROBLEM SETTING

In cross-lingual in-context learning, the prompt comprises $k$ rich-resource language demonstrations $(X_1, Y_1), (X_2, Y_2), ..., (X_k, Y_k)$ and a low-resource target language test input $X$, and the gold truth is $Y \in \mathbf{Y}$. For the generation-form task based on decoder-only LLMs, $\mathbf{Y}$ is the space of all possible token sequences. Similar to that of the topic model, a simplified assumption can be made for LLM

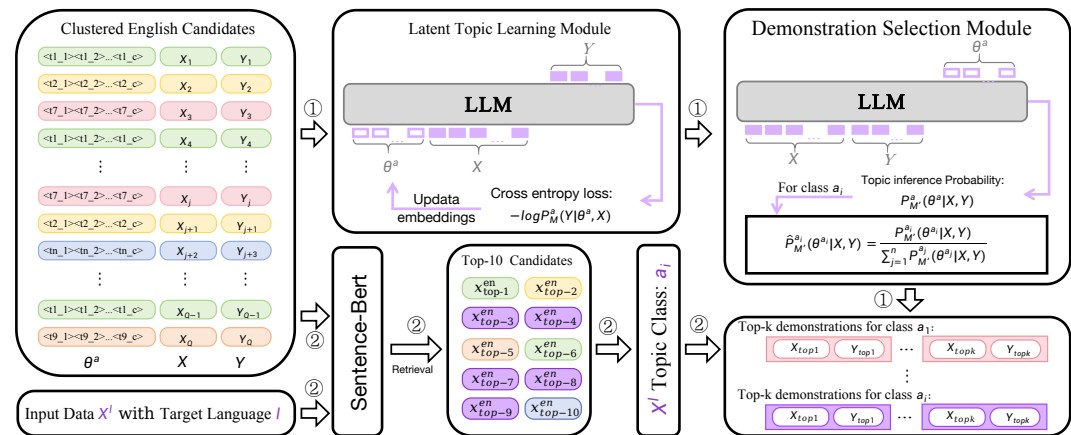

Figure 2: An overview of our proposed cross-lingual demonstration selection framework with topic inference.① Latent topic embeddings are learned for the clustered English candidates using LLMs, and probabilities of inferring to $n$ topics are calculated for each candidate. The top-k representative demonstrations for each topic are then obtained. ② For each target input, the semantic relationship with the candidates is calculated. The most frequent topic in the top-10 examples is used as its classification topic, denoted as $a_i$. The k most representative examples in the $a_i$ topic are used as the context for the target input, which can be used for ICL in any generative LLM.

(denoted by $M$):

$$P_M(Y|X) = \int_\Theta P_M(Y|\theta)P_M(\theta|X)d\theta, \tag{1}$$

$\theta \in \Theta$ is a high dimensional latent topic variable continuously distributed over $\Theta$, where $\Theta$ is the space of the variable.

Following Wang et al. (2023), we posit the existence of an underlying causal relation between $X, Y$, and $\theta$, directly named as $X \to Y \leftarrow \theta$, which can be represented mathematically as the following structural equation:

$$Y^a = \mathrm{f}(X^a, \theta^a, \epsilon), \tag{2}$$

where $\epsilon$ is an independent noise variable. $a$ is the topic of $(X, Y)$, and $\theta^a \in \Theta$ is the value of the topic variable corresponding to the topic $a$. The in-context learning output probability of LLM for an input $X^{a,l}$ classified to $a$ topic in target language $l$ can be denoted by $P_M^{a,l}$, and the solution can be defined as:

$$\arg\max_{y \in \mathbf{Y}} P_M^{a,l}(Y^{a,l} = y|X_1^a, Y_1^a, ..., X_k^a, Y_k^a, X^{a,l}). \tag{3}$$

It is always lower or equal to the Bayes optimal decoder:

$$\arg\max_{y \in \mathbf{Y}} P_M^{a,l}(Y^{a,l} = y|\theta^a, X^{a,l}).$$

Equality only holds when

$$P_M^{a,l}(\theta^a|X_1^a, Y_1^a, ..., X_k^a, Y_k^a, X^{a,l}) = 1 \tag{4}$$

Following Wang et al. (2023), we focus on estimating an optimal value of $\theta$ corresponding to a topic $a$. Then, we will discuss how to select an optimal set of demonstrations by using the learned optimal latent concept variable value.

## 3.2 LATENT TOPIC LEARNING

As shown in Fig.2, we first cluster the source language task dataset into several topics $\{a_i|i = 1, 2, ..., n\}$ by the multilingual embedding with K-means algorithm, the number of topic $n$ is a hyper-parameter. For a topic $a_i$, the objection of Bayes optimal decoder is to minimize $\mathbb{E}_{X,Y,a_i}[-\log P_M^{a_i}(Y|\theta^{a_i}, X)]$.

In practice, we try to align $\theta^a$ to the token embedding space by adding new tokens to the vocabulary of LLM. Then, the learned new tokens of $\theta^a$ are used as regular tokens in the vocabulary. Specifically, to represent each specific topic $a_i$, $c$ new topical tokens (denoted as $\hat{\theta}^{a_i}$) are added to the original vocabulary. $c$ is also a hyper-parameter, and corresponding $c$ topical tokens are appended to the input X as demonstrated, like "<t1_1><t1_2>...<t1_c>X" for the topic $a_1$. The new topical token can be anything as long as it does not overlap with the original vocabulary of LLM.

Subsequently, the embedding of these new tokens $E(\hat{\theta}^{a_i})$ is fine-tuned while freezing the remaining parameters of LLM. The fine-tuning objective is to minimize loss:

$$\mathcal{L}(\hat{\theta}^{a_i}) = \mathbb{E}_{X,Y}[-\log P_M^{a_i}(Y|\hat{\theta}^{a_i}, X)] \tag{5}$$

and the fine-tuned LLM denoted as $M'$. To obtain the topical tokens for all topics in a task, we fine-tune all data together with the loss $\sum_{i=1}^n \mathcal{L}(\hat{\theta}^{a_i})$.

### 3.3 DEMONSTRATION SELECTION

About the topic of target instance $(X^l, Y^l)$, we embed the input $X^l$ and measured its semantic similarity with all source input embeddings by Sentence-BERT (Reimers & Gurevych, 2019). Then, we statistic the topic category of the top-10 semantic similar source examples and choose the most frequent topic as the target language topic $a$.

According to the analysis in Section 3.1, for the target instances with topic $a$, our goal becomes selecting demonstrations that can best infer the topic for all inputs:

$$\underset{X_1^a, Y_1^a, ..., X_k^a, Y_k^a}{\arg\max} \mathbb{E}_X[P_M^a(\theta^a|X_1^a, Y_1^a, ..., X_k^a, Y_k^a, X)] \tag{6}$$

As test examples are sampled independently of the demonstrations and each demonstration is also sampled independently, the goal can be:

$$\underset{X_1^a, Y_1^a, ..., X_k^a, Y_k^a}{\arg\max} P_M^a(\theta^a|X_1^a, Y_1^a, ..., X_k^a, Y_k^a)$$
$$= \frac{\prod_{i=1}^k P_M^a(\theta^a|X_i^a, Y_i^a)}{P_M^a(\theta^a)^{k-1}} \tag{7}$$

Assuming that $\theta$ has a uniform prior, then our goal becomes finding the top $k$ demonstrations that maximize $\hat{P}_{M'}^a(\hat{\theta}^a|X_i^a, Y_i^a)$.

For the setting of $n$, the estimated conditional probability of $\hat{\theta}^{a_i}$ for instance $(X, Y)$ would be:

$$\hat{P}_{M'}^{a_i}(\hat{\theta}^{a_i}|(X, Y)) = \frac{P_{M'}^{a_i}(\hat{\theta}^{a_i}|(X, Y))}{\sum_{j=1}^n P_{M'}^{a_j}(\hat{\theta}^{a_j}|(X, Y))} \tag{8}$$

We mainly focus on the fundamental effects of topic inference on multilingual demonstration selection, without discussion of the mutual influence between demonstrations and the impact of order.

## 4 EXPERIMENTS

### 4.1 DATASET

This paper presents experiments conducted on three datasets: XNLI (Conneau et al., 2018), XCOPA[1], and TyDiQA-GoldP (Clark et al., 2020). The Cross-lingual Natural Language Inference dataset (XNLI) is a **sentence-pair classification** task involving 15 languages, translated from the English SNLI (Bowman et al., 2015) dataset. Since existing work mainly discusses demonstration selection methods for classification tasks, we also explored the multilingual **causal commonsense reasoning** task XCOPA and the **Question Answering** (QA) task in our experiments. XCOPA is an extension and re-annotation of the English Choice of Plausible Alternatives (COPA) dataset (Gordon

---

[1]https://github.com/cambridgeltl/xcopa

et al., 2012), with validation and test examples translated and annotated in 11 typologically diverse languages. TyDiQA-GoldP is the gold passage task in TyDiQA (Clark et al., 2020), covering 9 typologically diverse languages and serving as a challenging multilingual QA benchmark.

For each dataset, the English training set $\mathcal{D}$ serves as the pool of candidate demonstrations, evaluated across all test sets in each language. We list the English training set volume, 24 target languages, and their test set sizes in Table 3. The XCOPA test set is a combination of the official open-source 100 validation sets and 400 test sets. Due to the large size of the XNLI training dataset (392,701 instances in total), we only used the first 10,000 instances.

## 4.2 EXPERIMENTAL SETTING

We employ the K-means algorithm with random initial center points to cluster the training set $\mathcal{D}$, using three seed values $[32, 44, 100]$ and reporting the average results and standard deviation per language for $k = [2, 3, 4]$. Each training data representation is obtained using multilingual Sentence-BERT[2]. As for hyper-parameters, the number of cluster classes $n = 20$ and the length of each topic token sequence $c = 10$ are used for XNLI, and $n = 20$ and $c = 15$ are for TyDiQA-Gold, while $n = 5$ and $c = 15$ are set for XCOPA (with only 500 English training dataset). The guidelines for the hyper-parameters section can be seen in A.

We leverage the Bloomz-1b7[3] model to learn the topic token embeddings and compute the probability of each candidate. BLOOMZ-1b7 (Muennighoff et al., 2023) is a multilingual supervised fine-tuning version of BLOOM, which may be more efficient for learning the topic of a task. Greedy Search is employed for decoding answers in each task. For XCOPA, the gold output is changed to "1" or "2". For two-sentence tasks, we set the output length to 1 to obtain the answer label. For the QA task, the maximum output length is 16 and the answer is extracted by regular matching, and the metric is an Exact Match (EM) score. The prompts used for each task are detailed in Appendix B.

## 4.3 BASELINES

We use the same set of demonstrations for three LLMs, each with about 7 billion parameters, including BLOOM(Scao et al., 2022), Qwen1.5(Team, 2024), and Llama3.1(Dubey et al., 2024). We consider the following demonstration selection methods as baselines:

**ICL_random:** Random select $k$ demonstrations from $\mathcal{D}$ for each test example. We also set three seeds to obtain the average results.

**ICL_sem:** We use the same sentence-BERT to calculate the cosine similarity between the inputs of the source and target language. We select the top $k$ demonstrations from $\mathcal{D}$ for each test example.

**ICL_cluster:** Since our method first clusters $\mathcal{D}$ and then selects demonstrations, for ICL_cluster, we randomly sample $k$ instances from each category of the clustered data as demonstrations for all test examples within that category. The topic classification method for the test set follows the same procedure described in Section 3.3.

## 4.4 MAIN RESULTS

Table 1 presents our main results for the three datasets, averaged over all languages and based on the three LLMs. Across all three datasets, our method consistently outperforms the baselines across the three models. Figure 3 illustrates the performance difference between Topic-XICL and the best baseline results for individual low-resource languages in three LLMs across the three datasets. Languages marked with an asterisk (*) denote unseen languages for the models. Please refer to Appendix C for the definitions of the languages.

As shown in Table 1, **our method outperforms the strongest baseline across all three datasets and models in average performance**. For all $k$-value settings across the models, the average performance on the TyDIQA-GoldP, XCOPA, and XNLI tasks exceeds the strongest baseline by 3.32%, 2.47%, and 1.77%, respectively. For the MRC task, with $k = 4$, our method improves the EM

---

[2]https://huggingface.co/sentence-transformers/distiluse-base-multilingual-cased-v1
[3]https://huggingface.co/bigscience/bloomz-1b7

| Model | Method | TidyQA-GoldP (EM, %) | | | XCOPA (Accuracy, %) | | | XNLI (Accuracy, %) | | |
|---|---|---|---|---|---|---|---|---|---|---|
| | | k=2 | k=3 | k=4 | k=2 | k=3 | k=4 | k=2 | k=3 | k=4 |
| Qwen1.5 | Zero-shot | | 45.8 | | | 57.9 | | | 46.6 | |
| | ICL_random | 51.9±0.49 | 50.2±2.29 | 52.7±0.82 | 59.8±0.58 | 63.9±1.01 | 64.3±0.68 | 48.2±2.93 | 47.3±2.22 | 47.2±0.44 |
| | ICL_sem | 54.7 | 54.5 | 54.0 | 61.5 | 63.1 | 63.2 | 48.6 | 48.6 | 48.2 |
| | ICL_cluster | 53.0±0.55 | 53.1±0.49 | 53.2±0.45 | 61.1±0.64 | 63.4±0.70 | 64.1±0.81 | 48.6±0.31 | 48.3±0.22 | 47.6±0.36 |
| | Topic-XICL(ours) | **57.3±0.55** | **58.6±2.65** | **58.5±1.73** | **64.6±2.36** | **66.9±0.96** | **67.1±0.09** | **50.1±0.25** | **50.1±0.26** | **50.1±0.22** |
| BLOOM | Zero-shot | | 40.1 | | | 49.6 | | | 32.8 | |
| | ICL_random | 45.0±1.39 | 43.8±3.03 | 44.7±3.32 | 51.3±0.4 | 51.4±0.21 | 51.3±0.29 | 35.3±2.15 | 34.8±1.56 | 34.3±1.13 |
| | ICL_sem | 44.6 | 45.6 | 45.1 | 50.8 | 50.4 | 51.5 | 36.6 | 36.9 | 37.2 |
| | ICL_cluster | 45.6±0.97 | 45.1±1.26 | 45.0±0.88 | 51.7±0.09 | 51.0±0.26 | 51.9±0.17 | 34.4±0.92 | 35.2±1.67 | 36.1±1.59 |
| | Topic-XICL(ours) | **49.0±1.11** | **48.3±0.93** | **49.4±1.32** | **53.9±0.13** | **54.5±0.09** | **54.4±0.16** | **38.1±0.54** | **38.5±0.65** | **39.0±0.91** |
| Llama3.1 | Zero-shot | | 67.9 | | | 67.6 | | | 44.8 | |
| | ICL_random | 69.2±1.30 | 67.3±2.22 | 68.2±3.08 | 72.9±1.50 | 73.1±1.46 | 73.3±1.00 | 41.9±5.16 | 53.4±2.12 | 51.2±1.60 |
| | ICL_sem | 69.1 | 68.8 | 69.6 | 71.5 | 72.2 | 72.3 | 51.8 | 53.0 | 53.2 |
| | ICL_cluster | 69.6±0.34 | 69.2±0.23 | 69.6±0.54 | 72.9±0.25 | 73.4±0.25 | 73.4±0.48 | 51.1±1.08 | 52.3±0.61 | 53.1±0.66 |
| | Topic-XICL(ours) | **72.3±0.69** | **72.7±0.28** | **71.7±0.32** | **74.7±0.52** | **75.0±0.41** | **75.5±0.52** | **54.4±0.80** | **55.3±0.17** | **54.8±0.13** |

Table 1: Average performance and standard deviation over 3 seeds across languages for three tasks with different numbers of demonstrations. Detailed results in each language can be found in Appendix E.

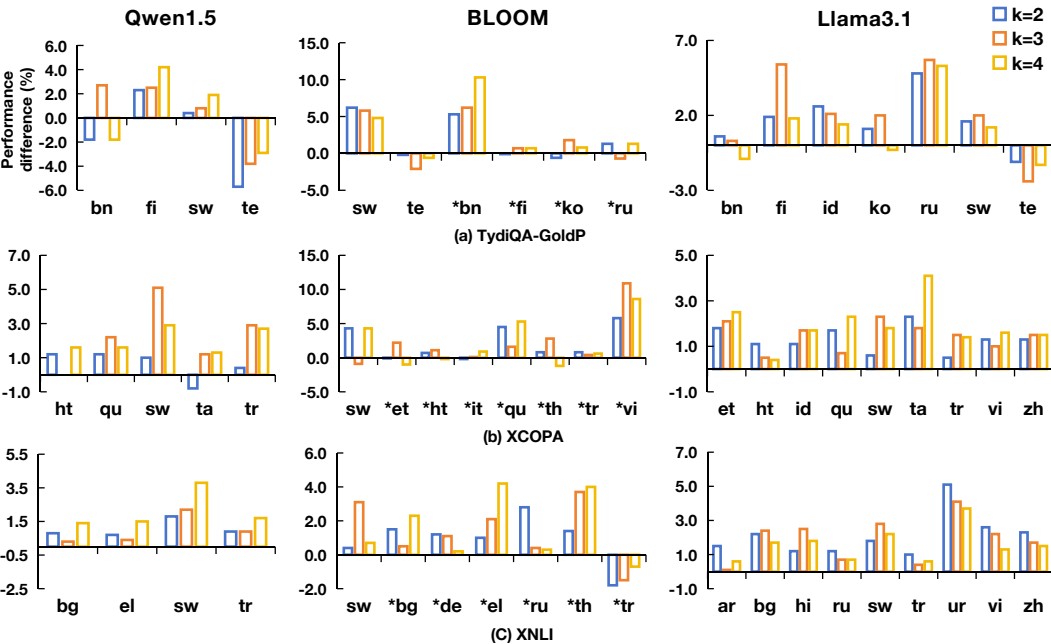

Figure 3: Performance difference between Topic-XICL and best baseline results for individual languages in Three datasets."*" represents the language is unseen for the models

score by 4.5% over the strongest baseline ICL_cluster on the BLOOM model. Furthermore, under different seed settings, our method achieves a smaller standard deviation compared to the random sampling method. This may be attributed to the clustering we performed on the dataset to obtain topics, as similar stability in performance across different seed settings can also be observed in the ICL_cluster method. ICL_cluster is a strong baseline that combines both semantic similarity and diversity. However, our method demonstrates an even greater advantage, suggesting that it not only benefits from the semantic similarity factors captured through clustering but also learns additional features automatically through the Topic variables.

In cross-lingual ICL settings, low-resource languages can achieve improvements with lightweight computational costs. **Our method demonstrates significant advantages in languages with relatively less training data (low-resource languages) or unseen languages across various models.** In the XNLI task, apart from BLOOM's performance on the unseen language Turkish (tr), where it did not surpass the strongest baseline, our method consistently outperforms existing baselines across all three models. Notably, our method achieves improvements of 4.2% and 4.0% in the unseen lan-

guages Greek (el) and Thai (th), respectively, over the best baseline on the BLOOM model with $k = 4$.

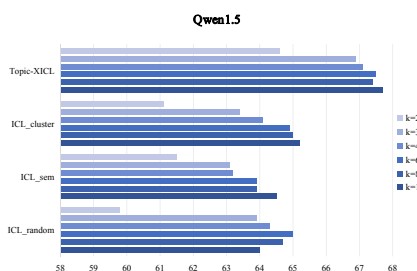

Figure 4: The performance of Qwen1.5 in XCOPA with $k = [2, 3, 4, 6, 8, 12]$.

In XCOPA, our method consistently outperforms the strongest baselines ICL_cluster. Specifically, it shows significant gains in low-resource languages, achieving a 10.9% improvement in the unseen language Vietnamese (vi) compared to ICL_cluster based on BLOOM. During our error analysis, we found that the simplicity of the XCOPA task (with '1' and '2' as the only possible answers) sometimes led baselines to overfit when there were too few examples and the labels were the same, causing the prediction to follow the answers from the demonstrations. To address this, we conducted experiments with larger $k$ values, e.g. $[6, 8, 12]$. The results for Qwen1.5 are shown in Figure 4 (other results are in Appendix E). Almost all methods show improvement as the value of $k$ increases, and our method maintains its advantage.

For more complex QA tasks like TyDiQA-GoldP, our model demonstrates significant advantages in certain low-resource languages. For instance, in Finnish (fi) with a $k = 4$ setting, our method surpasses the strongest ICL_sem baseline on Qwen1.5 by 4.2%, achieving a 64.7% EM score. In BLOOM, our best result in the unseen language Bengali (bn) exceeds the strongest baseline by 10.3%, reaching a 71.7% EM score. As for Llama3.1, this advantage may be attributed to its training on multilingual data (despite multilingual tokens making up only 8%), which gives it an edge in more low-resource languages.

Experimental results show that training the topic model on BLOOMZ-1b7 and selecting appropriate demonstrations enhances performance across different LLMs. At the task level, our method achieves notable improvements in more complex reasoning and question-answering tasks, demonstrating that our approach effectively applies Bayesian theory to non-classification tasks in ICL.

## 4.5 ABLATION STUDY

To validate the necessity of the topic model for demonstration selection and test data topic classification (as described in Section 3.3), we conducted ablation experiments. In the ICL_cluster, demonstration selection was replaced with random sampling. Our method calculates semantic similarity between the test data and candidate examples, selecting the most frequent topic from the top 10 similar candidate examples as its topic. We compared this with two simpler approaches: (1) using the topic of the most similar candidate example (top-1 topic) and (2) predicting the topic with a k-means clustering model (k-means predict). The results in Figure 5 show that the simpler classification methods outperformed the ICL_cluster

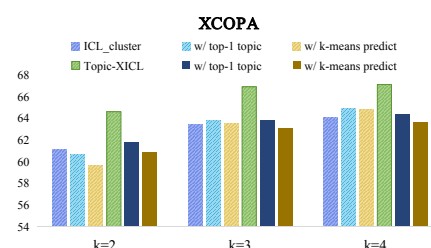

Figure 5: The ablation study in XCOPA based on Qwen1.5.

baseline, likely due to their effective matching of semantic similarities. However, they did not capture the broader characteristics of the entire candidate pool, limiting their effectiveness in our topic inference-based selection. In contrast, our approach consistently achieved the best results, demonstrating the value of the topic model and the necessity of each module.

## 5 ANALYSIS

The experimental results demonstrate that our topic model effectively captures latent information beneficial for in-context learning. We visualized the embeddings of the topic tokens to understand the relationships between different categories. Through case studies, we analyzed the characteristics of representative demonstrations within each topic. Additionally, we explored the performance of our method concerning model scale and source language.

## 5.1 VISUALIZATION OF TOPIC TOKEN EMBEDDING

As shown in Figure 6, the embeddings of the 20 topics trained on the TyDiQA-GoldP dataset are distributed into three to four distinct regions. This distribution indicates that the topic model successfully captures similarities across different topics. For instance, while topics "t10," "t11," and "t12" are part of separate clusters, they remain close in token sequence space. This demonstrates that even if initial clustering lacks precision, the topic model effectively identifies and groups similar topics. As a result, our method adapts well to different seed settings during initial clustering, resulting in lower standard deviations. For non-classification tasks, where topic classification may be ambiguous, our method displays strong adaptability. This highlights the broader applicability of our framework, extending the use of Bayesian theory in context sample selection to a wider range of tasks.

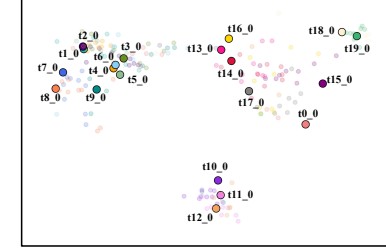

Figure 6: t-SNE plot of the learned topic tokens for TyDiQA-GoldP. "tx_0" represents the first token of the $x$th topic.

## 5.2 CASE STUDY FOR TOPIC-BASED DEMONSTRATION SELECTION

We examined representative examples from various topics in TyDiQA-GoldP, with detailed cases presented in Appendix D. Examples from topic "t0" mainly consist of paragraphs introducing items or concepts, often related to biology or species. Topic "t4" features shorter passages with only a few sentences, while topic "t18" focuses on sports and football themes. These examples demonstrate that our topic inference method not only captures simple semantic similarity but also structural and domain-level information.

## 5.3 RESULTS WITH LESS PARAMETER TOPIC MODEL

Considering that the boundaries between clusters can often be unclear when grouping source language candidate examples, we primarily trained our Topic model on BLOOMZ with 1.7 billion parameters. To verify the applicability of our method to smaller models, we also conducted experiments on BLOOMZ with 560 million parameters (BLOOMZ-560m). Figure 7 presents the ICL test results for three datasets on BLOOM with $k = 4$. Our method, when implemented on BLOOMZ-560m, continues to outperform the strongest baselines on XCOPA and TyDiQA-GoldP. Notably, for the MRC task in the Qwen1.5 model on the Arabic (ar) language, our method achieved an EM score of 65.8, exceeding the strongest baseline by 13.4%. However, in the XNLI task, only Vietnamese (vi) and Hindi (hi) languages showed improvements compared to the Topic model based on BLOOMZ-1b7, while other languages fell below the strongest baseline. This suggests that for simpler classification tasks, more clarity clustering information may be necessary.

In terms of time and resource consumption, given the similarity in training parameter size, training times for BLOOMZ-560m and

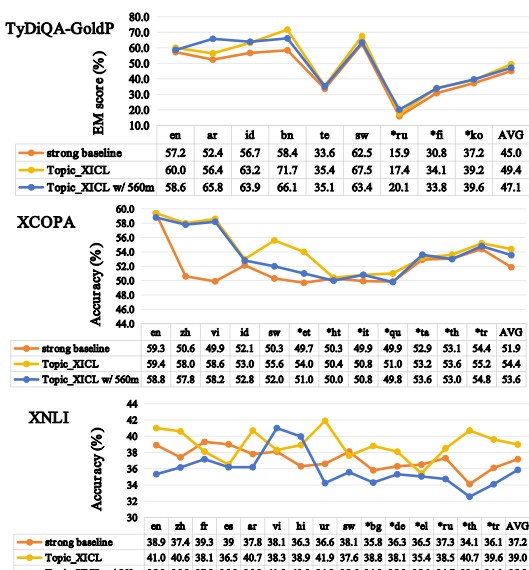

Figure 7: The 4-shot performance of BLOOM in three tasks based on the Topic-XICL model trained with fewer parameters (BLOOMZ-560m).

BLOOMZ-1b7 are approximately the same, taking only 15-30 minutes. Thus, for more complex reasoning or reading comprehension tasks, our method can leverage a smaller LLM while achieving performance improvements with minimal additional cost.

## 5.4 RESULTS WITH OTHER SOURCE LANGUAGES

For multilingual LLMs, besides English, other languages like Chinese and Italian also have relatively rich pre-training data. To explore how our method performs with different source languages, we conducted experiments using these two languages as the source languages for implementing Topic-XICL. We translated the English XCOPA training data into Chinese and Italian using the Google Translation API. The average results are shown in Table 9.

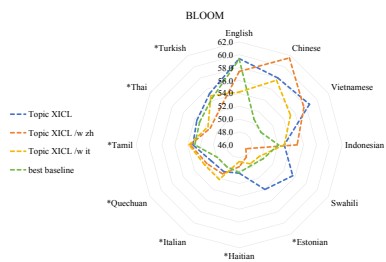

Figure 8: Results of 4-shot ICL for Individual Languages in XCOPA based on BLOOM by the Topic-XICL model trained with Chinese and Italian.

| Model | method | k=2 | k=3 | k=4 |
|---|---|---|---|---|
| Qwen1.5 | best baseline | 61.5 | 63.9 | 64.3 |
| | Topic XICL | 64.6 | **66.9** | **67.1** |
| | Topic XICL /w zh | 59.4 | 62.1 | 63.5 |
| | Topic XICL /w it | **65.3** | 65.7 | 64.4 |
| BLOOM | best baseline | 51.7 | 51.4 | 51.9 |
| | Topic XICL | **53.9** | **54.5** | **54.4** |
| | Topic XICL /w zh | 53.0 | 52.9 | 53.0 |
| | Topic XICL /w it | 52.4 | 52.8 | 52.7 |
| Llama3.1 | best baseline | 72.9 | 73.4 | 73.4 |
| | Topic XICL | **74.7** | 75.0 | 75.5 |
| | Topic XICL /w zh | 74.6 | 74.8 | 74.6 |
| | Topic XICL /w it | 74.5 | **75.6** | **75.6** |

Figure 9: The average accuracy of the Topic-XICL model trained with Chinese and Italian.

In both BLOOM and Llama3.1, the ICL performance of Topic-XICL demonstrations based on Chinese and Italian consistently outperforms the strongest baselines. On BLOOM, the non-Latin script Indonesian (id) language showed significant improvement, outperforming the English-based method by 2%, as shown in Figure 8. Moreover, in the Llama3.1 model, the average performance of Topic-XICL using Italian as the source language was even better than the English-based baseline. The improvements in Llama3.1 may stem from its better performance in relatively well-resourced target languages. For the Qwen1.5 model, which has more Chinese pre-training data, using Italian as the source language resulted in improvements over the baseline, but it failed to surpass the performance of the baseline when using Chinese. This could be because the stronger Chinese capabilities reduced alignment with other languages, making it harder to transfer knowledge effectively across languages.

English and Italian consistently achieve good XICL results across all models, but Chinese performs poorly as a source language for XICL in Qwen1.5. Across different target languages, there is no clear conclusion as to which source language's demonstrations provide more benefit. Zhang et al. (2024) conducted a multidimensional study on ICL for low-resource languages and found that the effectiveness of demonstration samples varies significantly across different models, tasks, and languages, which aligns with our conclusions. They also found that carefully designed templates can sometimes entirely negate the benefits of demonstration samples for certain tasks and languages. In our experiments, we also observed that for some languages, adjusting the prompt could yield greater improvements than ICL itself. However, this phenomenon is not consistent across all languages, posing a challenge for automatic multilingual prompt design. Our primary focus is on comparing the performance of ICL sample selection, while prompt selection will be explored in future work.

## 6 CONCLUSION

This work presents Topic-XICL, a novel demonstration selection algorithm for cross-lingual in-context learning that leverages latent topic inference. By integrating richer diversity information from latent topic variables based on a compact LLM, our method addresses the limitations of traditional similarity-based and task-specific retrievers. We validate its effectiveness across three task categories and three models, particularly in low-resource languages, demonstrating that latent topic variables effectively capture valuable diversity information for cross-lingual in-context learning. This approach provides a generalizable framework for enhancing XICL with only moderate additional resources.

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

## A EMPIRICAL GUIDELINES FOR HYPER-PARAMETER SELECTION

Regarding the choice of the number of topics (n) and tokens (c), there are empirical guidelines. For tasks with a large amount of English candidate data (greater than or equal to 2000), the number of clustering categories is set to $n = 20$, and for tasks with other data sizes, it is selected from (5, 10, 15), such as XCOPA with only 500 training data, which chooses $n = 5$. As for the topic tag sequence length, it is set to $c = 10$ for general classification tasks, and $c = 15$ for tasks that require reasoning or understanding of longer texts.

## B PROMPT TEMPLATE

Table 2 shows the prompt template we used for three tasks. For the TyDiQA-GoldP task with Qwen1.5, we translated the task description and the prompt 'Based on the passage, the answer to the question is' into the corresponding target language.

## C LOW-RESOURCE LANGUAGES

All 24 languages in the three datasets are not always pre-trained on the three baseline LLMs. Based on the language distribution in the pre-training data for each model, we selected some languages as low-resource or unseen languages, as shown in Table 4. For BLOOM (Scao et al., 2022), English training data accounts for 30.4% of the total, with pre-training data covering 46 natural languages. We define languages accounting for less than 0.1% as low-resource languages, and languages without training data are unseen languages. In Qwen 1.5 (Team, 2024), which boasts 7 billion parameters, the model supports a diverse set of 12 languages from Europe, East Asia, and Southeast Asia. We classify languages not included in their support as low-resource languages, as there is no clear distribution across all languages, making it difficult to define extremely low-resource languages.

Llama 3.1 (Dubey et al., 2024) supports English, German, French, Italian, Portuguese, Hindi, Spanish, and Thai. Other languages outside this list are considered low-resource languages, as Llama 3.1 uses about 8% multilingual tokens in their pre-train data (across 176 languages).

| Dataset | Prompt |
|---|---|
| **XNLI** | \<premise\>
Question: \<hypothesis\> True, False, or Inconclusive?
Answer: [True/False/Inconclusive] |
| **XCOPA** | Question: What might be the cause of / What might have happened as a result of "\<premise\>"?
Options:
1-\<Choice1\>
2-\<Choice2\>
You should tell me the choice number 1 or 2.
Answer: [1/2] |
| **TyDiQA-GoldP** | Answer the question from the given passage. Your answer should be directly extracted from the passage, and it should be a single entity, name, or number, not a sentence.
Passage: \<passage\>
question: \<question\>
Answer: Based on the passage, the answer to the question is "[a span in passage]" |

Table 2: Prompt template for three tasks.

| Dataset | Task | Languages | Train num. | Dev num. |
|---|---|---|---|---|
| XNLI | natural language inference | English(en), German(de), Russian(ru), French(fr), Spanish(es), Chinese(zh), Vietnamese(vi), Turkish(tr), Arabic(ar), Greek(el), Thai(th), Bulgarian(bg), Hindi(hi), Urdu(ur), Swahili(sw) | 10,000 | 5010 |
| XCOPA | commonsense reasoning | Chinese(zh), Italian(it), Vietnamese(vi), Indonesian(id), Turkish(tr), Thai(th), Estonian(es), Tamil(ta), Swahili(sw), Haitian(ht), Quechua(qu) | 500 | 500 |
| TyDiQA-GoldP | TyDiQA-GoldP | English(en), Russian(ru), Indonesian(id), Korean(ko), Arabic(ar), Finnish(fi), Bengali(bn), Telugu(te), Swahili(sw) | 3,695 | 113-921 |

Table 3: The detailed information of datasets.

| Model | Dataset | low-resource languages | extremly low-resource languages |
|---|---|---|---|
| BLOOM | XNLI | Swahili(sw) | German(de), Russian(ru), Turkish(tr), Greek(el), Thai(th), Bulgarian(bg) |
| | XCOPA | Swahili(sw) | Italian(it), Turkish(tr), Thai(th), Estonian(et), Haitian(ht), Quechua(qu) |
| | TyDiQA-GoldP | Telugu(te), Swahili(sw) | Russian(ru), Korean(ko), Finnish(fi), Bengali(bn) |
| Qwen1.5 | XNLI | Turkish(tr), Greek(el), Bulgarian(bg), Hindi(hi), Urdu(ur), Swahili(sw) | — |
| | XCOPA | Turkish(tr), Estonian(et), Tamil(ta), Swahili(sw), Haitian(ht), Quechua(qu) | — |
| | TyDiQA-GoldP | Finnish(fi), Bengali(bn), Telugu(te), Swahili(sw) | — |
| Llama3.1 | XNLI | Russian(ru), Chinese(zh), Vietnamese(vi), Turkish(tr), Arabic(ar), Greek(el), Bulgarian(bg), Hindi(hi), Urdu(ur), Swahili(sw) | — |
| | XCOPA | Chinese(zh), Vietnamese(vi), Indonesian(id), Turkish(tr), Estonian(et), Tamil(ta), Swahili(sw), Haitian(ht), Quechua(qu) | — |
| | TyDiQA-GoldP | Russian(ru), Indonesian(id), Korean(ko), Arabic(ar), Finnish(fi),Bengali(bn), Telugu(te), Swahili(sw) | — |

Table 4: Classification of languages for three datasets (XNLI, XCOPA, TyDiQA-GoldP) across three LLMs (BLOOM, Qwen1.5, Llama3.1).

## D  CASE STUDY

Table 5 shows the representative examples selected from some topics in TyDiQA-GoldP.

## E  DETAILED RESULTS

All results in individual languages of three tasks are reported in Tables 6, 7, and 8. For XCOPA, we report the results with large $k$ value settings in Figure 10.

| Topic | Top-4 Examples | | |
|---|---|---|---|
| | Passage | Question | Answer |
| t0 | **Brontosaurus** was a large, long-necked quadrupedal animal with a long, whip-like tail, and forelimbs that were slightly shorter than its hindlimbs. ... The largest species, B. excelsus, weighed up to 15 tonnes (15 long tons; 17 short tons) and measured up to 22m (72ft) long from head to tail... | How tall were brontosaurs? | 22m (72ft) |
| | **Pteranodon** was the first pterosaur found outside of Europe. Its fossils first were found by Othniel Charles Marsh in 1870,... | Where was the first Pteranodon fossil found? | Othniel Charles Marsh |
| | **Sawfishes**, also known as carpenter sharks, are a family of rays characterized by a long, ... They are among the largest fish with some species reaching lengths of about 7–7.6m (23–25ft)... | How long do sawfishes get?? | 7–7.6m (23–25ft) |
| | The name "**Haflinger**" comes from the village of Hafling, which today is in northern Italy. ..The desired height today is between 13.2 and 15 hands (54 and 60 inches, 137 and 152 cm). ... | How tall does Haflingers get? | (54 and 60 inches) |
| t4 | HGTV (an initialism for Home & Garden Television) is an American basic cable and satellite television channel that is owned by Discovery, Inc. | Who owns HGTV? | Discovery, Inc |
| | The pope is the bishop of Rome. He is also, by virtue of that office: Vicar of Jesus Christ, Successor of the Prince of the Apostles, Supreme Pontiff of the Universal Church, Patriarch of the Latin Church, Primate of Italy, Archbishop and Metropolitan of the Roman Province, Sovereign of the Vatican City State, Servant of the servants of God. | Who runs the Catholic Church? | The pope |
| | The National Insurance number is a number used in the United Kingdom in the administration of the National Insurance or social security system. It is also used for some purposes in the UK tax system. The number is described by the United Kingdom government as a "personal account number." | What is the average age that people have their first child in the UK? | 27 to 29 years old |
| | The National Insurance number is a number used in the United Kingdom in the administration of the National Insurance or social security system. It is also used for some purposes in the UK tax system. The number is described by the United Kingdom government as a "personal account number." | What is the British equivalent of Social Security? | The National Insurance number |
| t18 | The 2009 **Stanley Cup Finals** was the championship series of the **National Hockey League's** (NHL) 2008–09 season, and the culmination of the 2009 **Stanley Cup playoffs**. It was contested between the Eastern Conference champion Pittsburgh Penguins and the Western Conference champion Detroit Red Wings. ... | Who won the 2009 Stanley Cup | Pittsburgh |
| | The history of the **Philadelphia Eagles** begins in 1933. In their history, the Eagles have appeared in the **Super Bowl** three times, losing in their first two appearances but winning the third, in 2018. They won three NFL Championships, the precursor to the Super Bowl, in four appearances... | How many times have the Philadelphia Eagles played in the Super Bowl? | three times |
| | The **Pittsburgh Steelers** (6–2) have won the most **Super Bowls** with six championships, while the New England Patriots (5–5), the Dallas Cowboys (5–3), and the San Francisco 49ers (5–1) have five wins. New England has the most Super Bowl appearances with eleven, while the Buffalo Bills (0–4) ... | Who won the last Super Bowl? | New England Patriots |
| | Adam Matthew Vinatieri (born December 28, 1972) is an **American football placekicker** for the Indianapolis Colts of the **National Football League (NFL)**. He has played in five Super Bowls: four with the New England Patriots and one with the Colts, winning with the Patriots in 2001, 2003, and 2004 ... | Who is the oldest player in the NFL? | Vinatieri |

Table 5: The top-4 representative samples of some topics in TyDiQA-GoldP selected by our Topic-XICL model.

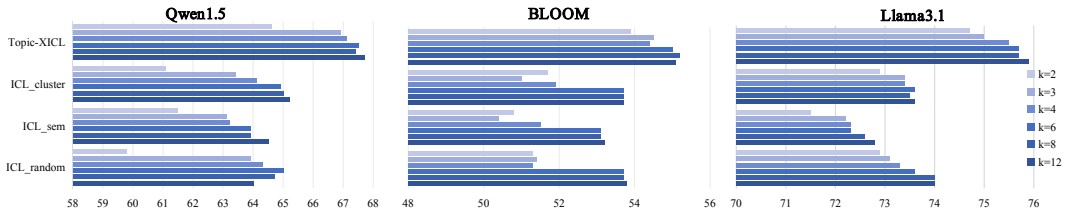

Figure 10: The performance of BLOOM-7b1, Qwen1.5-7B and Llama-3.1-8B models in XCOPA with $k = [2, 3, 4, 6, 8, 12]$.

| TyDiQA-GoldP | ar | bg | en | fi | id | ko | ru | sw | te | avg |
|---|---|---|---|---|---|---|---|---|---|---|
| **Qwen1.5-7B** | 44.3 | 46.9 | 62.3 | 53.2 | 63.0 | 52.5 | 43.2 | 37.3 | 9.1 | 45.8 |
| k=2 ICL_random | 42.1±0.41 | 37.5±1.10 | 63.9±0.74 | 56.8±0.80 | 64.5±0.87 | 62.1±0.95 | 46.4±0.46 | 59.7±2.30 | 33.9±2.26 | 51.9±0.49 |
| k=2 ICL_sem | 45.4 | **42.5** | 63.6 | 60.9 | 65.8 | 62.3 | 48.9 | 62.3 | **40.4** | 54.7 |
| k=2 ICL_cluster | 45.1±1.07 | 40.7±1.45 | 65.2±0.74 | 56.1±0.97 | 65.5±0.66 | 62.7±0.30 | 48.1±0.41 | 59.9±0.34 | 33.3±2.40 | 53.0±0.55 |
| k=2 Topic-XICL(ours) | **53.4±4.70** | 40.7±1.10 | **68.2±1.06** | **63.2±2.31** | **72.2±3.07** | **71.4±3.02** | **49.0±0.86** | **62.7±0.33** | 34.7±0.55 | **57.3±0.55** |
| k=3 ICL_random | 40.6±3.49 | 39.2±1.10 | 63.5±1.72 | 54.9±4.13 | 64.4±1.01 | 62.9±1.97 | 46.9±1.37 | 53.8±5.15 | 27.1±4.50 | 50.2±2.29 |
| k=3 ICL_sem | 44.8 | 40.7 | 63.6 | 61.3 | 65.8 | 64.1 | 47.4 | 61.9 | **40.4** | 54.5 |
| k=3 ICL_cluster | 45.0±0.58 | 40.1±0.42 | 65.6±0.54 | 56.9±2.27 | 64.6±1.15 | 62.8±2.10 | 48.1±0.61 | 59.3±2.69 | 35.2±1.31 | 53.1±0.49 |
| k=3 Topic-XICL(ours) | **54.0±5.12** | **43.4±1.50** | **69.8±1.93** | **63.8±2.29** | **73.3±3.52** | **74.6±3.97** | **49.4±1.01** | **62.7±1.18** | 34.0±0.26 | **58.6±2.65** |
| k=4 ICL_random | 42.6±1.38 | 37.8±1.10 | 65.2±1.03 | 58.9±1.39 | 65.5±1.28 | 62.4±1.52 | 47.3±1.16 | 58.4±2.20 | 36.0±0.97 | 52.7±0.82 |
| k=4 ICL_sem | 42.8 | **41.6** | 64.5 | 60.5 | 65.8 | 64.5 | 48.0 | 58.7 | **39.8** | 54.0 |
| k=4 ICL_cluster | 43.8±0.71 | 40.1±0.42 | 65.7±0.81 | 56.9±0.70 | 65.9±0.73 | 62.6±1.20 | 47.5±0.79 | 60.4±0.50 | 35.6±2.61 | 53.2±0.45 |
| k=4 Topic-XICL(ours) | **55.7±5.76** | 39.8±1.25 | **70.9±2.15** | **64.7±2.52** | **73.3±3.28** | **73.2±3.70** | **49.3±0.79** | **62.3±0.62** | 36.9±1.73 | **58.5±1.73** |
| **BLOOM-7b1** | 27.1 | 39.8 | 54.5 | **29.2** | 59.5 | 29.7 | 31.8 | 69.1 | 19.7 | 40.1 |
| k=2 ICL_random | 50.8±5.23 | 64.6±1.45 | 53.8±1.87 | 11.8±2.00 | 57.9±2.49 | **33.2±0.62** | 36.7±1.48 | 63.1±1.56 | 33.0±2.06 | 45.0±1.39 |
| k=2 ICL_sem | 52.8 | 57.5 | 55.5 | 18.8 | 56.1 | 28.6 | 36.0 | 61.3 | **35.0** | 44.6 |
| k=2 ICL_cluster | **56.5±3.20** | 60.8±2.21 | 54.7±0.84 | 15.5±2.22 | 59.4±0.80 | 31.6±0.34 | 37.7±0.86 | 62.6±2.62 | 31.1±2.21 | 45.6±0.97 |
| k=2 Topic-XICL(ours) | 56.0±1.08 | **69.9±1.45** | **56.1±0.57** | 18.7±0.88 | **64.1±1.89** | 32.6±0.17 | **39.0±0.95** | **69.3±2.78** | 34.8±1.11 | **49.0±1.11** |
| k=3 ICL_random | 48.2±6.54 | 61.9±4.74 | 53.9±4.84 | 14.0±2.77 | 55.6±4.59 | 31.2±2.63 | 35.3±3.23 | 61.3±3.03 | 32.9±3.03 | 43.8±3.03 |
| k=3 ICL_sem | 52.0 | 59.3 | **58.6** | 16.8 | 57.2 | 30.1 | **38.5** | 61.7 | **36.5** | 45.6 |
| k=3 ICL_cluster | 55.4±2.41 | 62.8±2.50 | 55.4±0.75 | 14.7±1.25 | 57.2±1.95 | 29.3±2.13 | 38.1±1.67 | 61.3±2.16 | 31.2±3.88 | 45.1±1.26 |
| k=3 Topic-XICL(ours) | **55.5±1.90** | **70.6±0.72** | 58.2±1.13 | 17.5±0.34 | **61.8±0.80** | **33.0±0.34** | 37.8±0.31 | **67.5±1.80** | 34.4±0.93 | **48.3±0.93** |
| k=4 ICL_random | 47.4±6.72 | 61.4±4.17 | 54.9±4.70 | 16.7±3.52 | 55.8±5.06 | 32.2±2.35 | 36.2±4.12 | 62.7±2.05 | 35.5±2.97 | 44.7±3.32 |
| k=4 ICL_sem | 48.9 | 57.5 | 58.2 | 16.2 | 57.5 | 33.3 | 37.9 | 60.7 | 36.0 | 45.1 |
| k=4 ICL_cluster | 52.4±1.93 | 58.4±1.91 | 57.2±1.09 | 15.9±2.08 | 56.7±1.03 | 30.8±1.57 | 37.2±1.92 | 62.5±0.75 | 33.6±3.81 | 45.0±0.88 |
| k=4 Topic-XICL(ours) | **56.4±1.30** | **71.7±2.32** | **60.0±0.70** | 17.4±0.89 | **63.2±0.60** | **34.1±0.17** | **39.2±0.27** | **67.5±1.58** | 35.4±1.32 | **49.4±1.32** |
| **Llama-3.1-8B** | 61.0 | 57.5 | 70.7 | 65.5 | 70.1 | 77.5 | 51.2 | 78.4 | 79.4 | 67.9 |
| k=2 ICL_random | 60.0±2.31 | 64.9±1.10 | 71.1±1.42 | 65.0±1.74 | 70.3±1.53 | 81.0±1.52 | 53.4±2.44 | 79.1±0.50 | 77.6±0.85 | 69.2±1.30 |
| k=2 ICL_sem | 59.7 | 61.1 | 71.4 | 66.4 | 69.4 | 81.5 | 52.6 | 80.4 | **80.0** | 69.1 |
| k=2 ICL_cluster | 61.2±1.22 | 66.7±1.82 | 71.3±0.28 | 65.9±0.91 | 68.8±0.38 | 80.2±0.62 | 54.2±1.31 | 79.9±0.57 | 77.9±0.92 | 69.6±0.34 |
| k=2 Topic-XICL(ours) | **65.4±2.69** | **67.3±2.50** | **74.3±1.61** | **68.3±1.05** | **72.9±0.83** | **82.6±0.00** | **59.4±2.69** | **82.0±0.66** | 78.9±0.69 | **72.3±0.69** |
| k=3 ICL_random | 57.4±3.90 | 62.8±3.15 | 68.6±2.19 | 62.4±2.90 | 67.9±1.31 | 81.3±1.40 | 52.1±2.85 | 76.6±0.66 | 76.4±2.24 | 67.3±2.22 |
| k=3 ICL_sem | 57.4 | 63.7 | 71.1 | 64.5 | 70.3 | 80.1 | 52.0 | 80.2 | **80.1** | 68.8 |
| k=3 ICL_cluster | 60.6±0.40 | 67.8±1.50 | 70.1±0.39 | 64.7±0.36 | 69.6±0.17 | 80.1±0.30 | 53.7±0.50 | 78.4±0.96 | 77.4±1.06 | 69.2±0.23 |
| k=3 Topic-XICL(ours) | **66.1±3.16** | **68.1±1.35** | **75.0±2.04** | **70.1±1.88** | **72.4±1.18** | **83.3±0.59** | **59.4±3.31** | **82.2±1.25** | 77.7±0.28 | **72.7±0.28** |
| k=4 ICL_random | 58.8±4.84 | 64.9±3.34 | 69.7±4.29 | 63.9±3.76 | 69.3±2.30 | 81.8±1.81 | 53.6±3.27 | 77.7±2.65 | 74.3±3.76 | 68.2±3.08 |
| k=4 ICL_sem | 57.7 | 66.4 | 71.6 | 66.4 | 70.3 | **82.2** | 53.8 | 80.2 | **78.3** | 69.6 |
| k=4 ICL_cluster | 60.3±0.85 | **69.0±1.91** | 70.5±0.57 | 66.5±0.49 | 69.6±0.60 | 80.1±0.78 | 53.8±1.16 | 79.2±0.81 | 77.4±1.29 | 69.6±0.54 |
| k=4 Topic-XICL(ours) | **64.6±2.45** | 68.1±1.45 | **73.6±1.24** | **68.3±0.85** | 71.7±0.66 | 81.9±0.34 | **59.1±3.14** | **81.4±1.23** | 77.0±0.32 | **71.7±0.32** |

Table 6: F1 score of TyDiQA-GoldP in 9 languages based on BLOOM-7b1, Qwen1.5-7B and Llama-3.1-8B models.

| XCOPA | en | ar | bg | de | el | es | fr | hi | ru | sw | th | tr | ur | vi | zh | AVG |
|---|---|---|---|---|---|---|---|---|---|---|---|---|---|---|---|---|
| **Qwen1.5-7B** | 58.5 | **49.1** | 41.1 | **53.4** | 46.0 | 52.2 | **54.5** | 42.3 | 44.1 | 36.6 | 37.8 | 44.7 | 39.7 | 50.4 | 49.4 | 46.6 |
| k=2 ICL_random | **63.1±5.19** | 45.6±3.22 | 48.5±3.16 | 50.0±3.33 | 49.2±3.58 | **52.3±3.90** | 49.6±3.69 | 45.9±1.37 | 48.6±2.83 | 35.4±0.99 | 46.6±3.48 | 43.0±1.51 | 40.8±1.81 | 45.6±4.28 | 53.7±2.44 | 48.2±2.93 |
| k=2 ICL_sem | 58.1 | 46.0 | 49.8 | 51.6 | 48.5 | 51.2 | 50.8 | 46.9 | 48.1 | 38.5 | 46.8 | 45.2 | 42.4 | 52.7 | 52.6 | 48.6 |
| k=2 ICL_cluster | 58.8±0.26 | 45.5±0.74 | 49.3±1.18 | 50.9±0.58 | 47.4±0.75 | 52.7±1.07 | 50.3±0.39 | 47.6±0.51 | 49.0±0.43 | 38.0±0.88 | 48.1±3.95 | 44.7±0.99 | 41.1±2.33 | 52.8±1.88 | 53.3±0.75 | 48.6±0.31 |
| k=2 Topic-XICL(ours) | 59.6±0.22 | 47.3±0.91 | **50.6±0.79** | 51.8±0.34 | **49.9±0.11** | 52.0±0.20 | 51.2±0.12 | **48.2±0.05** | **49.5±0.49** | **40.3±0.33** | **50.9±1.50** | **46.1±0.95** | **44.3±0.55** | **55.0±1.81** | **54.6±0.25** | **50.1±0.25** |
| k=3 ICL_random | 59.2±1.93 | 44.2±2.76 | 47.6±2.07 | 49.8±2.96 | 47.6±1.33 | 50.8±2.55 | 49.4±2.77 | 47.6±1.89 | 47.8±2.46 | 35.1±0.77 | 44.6±3.75 | 42.5±1.61 | 40.7±2.02 | 49.9±2.70 | 53.3±1.95 | 47.3±2.22 |
| k=3 ICL_sem | 56.7 | 46.3 | 49.5 | 51.0 | 49.5 | 50.8 | 50.7 | 48.0 | 48.7 | 37.4 | 47.6 | 45.0 | 42.3 | 52.5 | 52.0 | 48.6 |
| k=3 ICL_cluster | 58.2±0.75 | 45.1±0.49 | 48.4±0.92 | 50.4±0.42 | 48.3±2.01 | 51.4±0.91 | 49.9±0.27 | 48.8±0.09 | 47.6±0.73 | 38.6±1.95 | 48.1±2.71 | 44.7±0.99 | 41.4±2.38 | 52.4±2.39 | 51.6±0.66 | 48.3±0.22 |
| k=3 Topic-XICL(ours) | **59.9±0.45** | 46.7±0.81 | **49.8±0.74** | **52.4±0.45** | **49.9±0.49** | 51.9±0.56 | 48.7±0.18 | 49.4±0.20 | **40.1±1.60** | **50.1±1.18** | 45.9±0.77 | **45.0±0.82** | **54.6±1.07** | **54.0±0.26** | **50.1±0.26** | |
| k=4 ICL_random | **60.0±0.75** | 44.1±0.69 | 47.3±0.54 | 50.0±0.75 | 47.5±0.60 | 50.9±0.31 | 49.8±0.92 | 48.0±1.04 | 47.9±0.40 | 34.4±0.30 | 43.1±1.99 | 41.8±0.18 | 40.2±1.68 | 49.6±0.35 | **53.4±0.69** | 47.2±0.04 |
| k=4 ICL_sem | 56.1 | 45.4 | 48.6 | 50.9 | 49.2 | 50.5 | 50.2 | 48.1 | 48.7 | 36.7 | 47.2 | 34.7 | 42.9 | 52.2 | 51.7 | 48.2 |
| k=4 ICL_cluster | 58.4±0.44 | 44.8±0.82 | 47.1±0.64 | 50.6±0.44 | 47.6±2.18 | 50.6±0.93 | 49.7±0.14 | 48.1±0.44 | 47.4±0.85 | 36.9±2.45 | 46.4±2.31 | 43.5±0.91 | 41.6±3.15 | 50.8±3.16 | 50.6±0.54 | 47.6±0.36 |
| k=4 Topic-XICL(ours) | 59.1±0.52 | 47.3±1.14 | **50.0±1.35** | 51.4±0.48 | **50.7±1.33** | **52.8±0.84** | 51.5±0.81 | 48.3±0.05 | 49.2±0.87 | **40.7±2.65** | **50.1±1.15** | **46.4±1.30** | **45.1±1.44** | **55.1±1.44** | 53.0±0.22 | **50.1±0.22** |
| **BLOOM-7b1** | 34.1 | 33.6 | 33.7 | 33.1 | 33.4 | 35.8 | 36.5 | 31 | 33.4 | 32.9 | 21.2 | 33.6 | 33.3 | 33.1 | 32.7 | 32.8 |
| k=2 ICL_random | 37.8±5.13 | 35.1±2.47 | 34.7±1.49 | 34.5±0.94 | 34.5±0.27 | **38.3±5.14** | 37.9±5.62 | 33.8±0.65 | 34.0±0.42 | 36.2±2.65 | 34.9±2.34 | 34.6±1.58 | 34.4±1.51 | 34.6±1.52 | 34.2±1.33 | 35.3±2.15 |
| k=2 ICL_sem | 37.9 | 35.9 | 36.3 | 35.8 | 36.1 | 38 | 37.6 | 36.2 | 36.2 | 38.8 | 35.3 | **36.5** | 34.7 | 38.6 | 35.1 | 36.6 |
| k=2 ICL_cluster | 35.7±1.63 | 33.8±1.33 | 34.9±0.2 | 34.3±1.1 | 35.0±1.3 | 35.3±1.49 | 35.5±1.58 | 36.2±0.59 | 35.5±0.35 | 36.1±0.47 | 33.9±1.4 | 33.7±0.28 | 33.2±0.93 | 33.5±0.83 | 33.7±1.41 | 34.4±0.92 |
| k=2 Topic-XICL(ours) | **38.7±0.11** | **38.1±0.08** | **37.8±0.41** | **37.0±0.07** | 37.1±1.43 | 37.3±0.68 | 38.3±0.75 | **42.1±4.05** | **39.0±0.44** | **39.2±0.38** | **36.7±1.84** | 34.7±0.36 | **37.4±0.72** | **39.8±2.94** | **38.5±0.88** | **38.1±0.54** |
| k=3 ICL_random | 35.9±3.29 | 34.8±3.24 | 34.3±0.3 | 34.9±1.54 | 36.3±2.56 | 35.1±3.23 | 35.0±2.47 | 34.2±2.31 | 35.0±2.28 | 34.3±2.06 | 33.9±0.83 | 33.3±0.41 | 34.5±2.18 | 35.4±3.18 | 35.0±3.18 | 34.8±1.56 |
| k=3 ICL_sem | 38.3 | **37.6** | 36.7 | 35.7 | 36.6 | 37.6 | 37.7 | 36.4 | 37.3 | 37.7 | 34.1 | **36.6** | 36.2 | 38.1 | 36.2 | 36.9 |
| k=3 ICL_cluster | 36.4±1.38 | 35.6±2.53 | 35.7±1.54 | 35.2±0.95 | 36.6±0.49 | 36.3±1.16 | 34.9±2.77 | 35.8±0.67 | 36.1±0.55 | 38.0±1.46 | 33.4±0.94 | 34.1±2.57 | 33.8±2.16 | 33.8±2.19 | 35.2±1.67 | 35.2±1.67 |
| k=3 Topic-XICL(ours) | **41.1±0.69** | 35.2±0.82 | **37.2±0.09** | **36.8±0.42** | **38.7±1.88** | **39.8±0.61** | **39.9±0.41** | **41.5±2.30** | **37.7±0.38** | **41.1±1.55** | **37.8±1.57** | 35.1±0.34 | **37.8±0.91** | **39.8±1.85** | **38.7±0.24** | **38.5±0.65** |
| k=4 ICL_random | 33.6±2.17 | 34.9±2.34 | 33.4±0.64 | 35.0±0.82 | 33.4±0.53 | 33.1±0.72 | 33.5±0.91 | 35.1±2.63 | 35.8±0.95 | 34.6±0.76 | 33.3±0.38 | 33.2±0.61 | 34.1±1.19 | 35.2±3.53 | 35.3±2.39 | 34.3±1.13 |
| k=4 ICL_sem | 38.9 | 37.8 | 35.8 | 36.3 | 36.5 | 39 | **39.3** | 36.3 | 37.3 | 38.1 | 34.1 | 36.1 | 36.6 | 38.1 | 37.4 | 37.2 |
| k=4 ICL_cluster | 36.6±1.61 | 36.3±1.79 | 35.1±1.83 | 36.0±1.63 | 33.9±0.61 | 35.0±1.48 | 36.7±1.48 | 36.7±0.83 | 37.7±1.51 | 36.2±2.68 | 33.8±1.71 | 34.0±2.68 | 36.2±2.61 | 39.2±2.22 | 36.0±1.59 | 35.2±1.21 |
| k=4 Topic-XICL(ours) | **41.0±1.01** | **40.6±0.51** | **38.1±1.34** | **36.5±0.45** | **40.7±3.00** | 38.3±0.65 | 38.9±0.61 | **41.9±3.46** | 37.6±0.25 | **38.8±2.23** | **38.1±1.88** | 35.4±0.38 | **38.5±0.94** | **40.7±3.14** | **39.6±1.17** | **39.0±0.91** |
| **Llama-3.1-8B** | 52.3 | 45.5 | 45.7 | 41.4 | 51.4 | 50.9 | 44.0 | 46.2 | 38.6 | 46.6 | 44.5 | 31.3 | 47.3 | 49.4 | 44.8 | |
| k=2 ICL_random | 47.5±9.35 | 43.6±2.86 | 42.6±7.65 | 47.8±2.93 | 45.2±7.87 | 41.4±5.16 | 43.4±4.62 | 40.6±6.03 | 42.2±6.82 | 38.1±1.95 | 37.0±3.17 | 41.0±5.89 | 37.4±2.98 | 39.8±3.46 | 40.9±7.41 | 41.9±5.16 |
| k=2 ICL_sem | 60.5 | 51.6 | 52.3 | 54.0 | 52.2 | 54.6 | 54.8 | 48.8 | 53.0 | 46.5 | 47.9 | 51.1 | 46.0 | 51.9 | 53.4 | 51.8 |
| k=2 ICL_cluster | 58.9±1.86 | 51.7±0.95 | 52.4±2.52 | 54.1±0.82 | 44.7±7.43 | 55.8±0.84 | 53.2±3.22 | 51.4±2.63 | 51.5±2.03 | 44.3±0.83 | 47.0±3.46 | 51.6±2.10 | 44.5±4.76 | 53.2±0.34 | 51.9±0.71 | 51.1±1.08 |
| k=2 Topic-XICL(ours) | **61.3±0.20** | **53.2±0.77** | **54.6±0.18** | **55.8±0.83** | **56.1±0.30** | **56.3±1.14** | **56.4±1.11** | **52.6±0.11** | **54.2±0.27** | **48.3±0.93** | **52.7±1.95** | **52.6±0.64** | **51.1±2.48** | **55.8±0.78** | **54.5±0.80** | **54.4±0.80** |
| k=3 ICL_random | **64.0±2.49** | 53.5±2.11 | 54.3±1.95 | 56.3±1.78 | 52.0±2.80 | **58.0±4.59** | 57.4±4.31 | 51.1±2.62 | 53.1±2.56 | 46.2±2.90 | 48.2±3.76 | 53.4±2.38 | 46.3±1.55 | 54.4±4.52 | 52.5±3.06 | 53.4±2.12 |
| k=3 ICL_sem | 62.4 | 52.6 | 53.7 | 55.1 | 53.2 | 56.5 | 56.0 | 51.0 | 54.6 | 46.3 | 49.4 | 52.4 | 46.9 | 51.9 | 53.4 | 53.0 |
| k=3 ICL_cluster | 61.3±1.50 | 52.4±0.60 | 52.5±3.36 | 54.6±1.02 | 55.0±1.54 | 56.1±0.49 | 54.8±0.67 | 52.1±2.16 | 50.6±2.06 | 44.3±1.48 | 48.1±3.39 | 53.0±0.74 | 45.4±4.25 | 51.6±1.12 | 52.5±2.10 | 52.3±0.61 |
| k=3 Topic-XICL(ours) | 63.6±0.24 | **53.6±0.53** | **56.7±0.61** | **56.3±0.38** | **56.2±0.40** | 57.9±0.80 | 57.4±0.54 | **54.6±0.68** | **55.2±0.43** | **49.1±0.31** | **52.4±0.08** | **53.8±0.41** | **50.6±0.28** | **56.6±0.84** | **55.1±0.17** | **55.3±0.17** |
| k=4 ICL_random | 59.4±4.06 | 52.8±0.22 | 50.8±3.48 | **56.3±1.31** | 48.7±1.74 | 55.3±3.27 | **57.0±2.92** | 48.4±2.98 | 49.8±4.96 | 45.3±0.58 | 46.3±2.39 | 50.9±2.01 | 45.3±2.02 | 51.7±2.18 | 49.6±3.49 | 51.2±1.60 |
| k=4 ICL_sem | 62.9 | 52.6 | 54.0 | 54.8 | 53.3 | **56.4** | 56.0 | 51.2 | 45.9 | 50.0 | 52.6 | 46.7 | 51.8 | 53.1 | 53.2 | |
| k=4 ICL_cluster | 62.9±0.92 | 52.1±0.51 | 54.0±1.23 | 54.8±0.76 | 55.6±1.71 | 56.4±0.73 | 54.5±0.56 | 52.6±1.26 | 52.0±1.82 | 45.0±1.64 | 50.0±2.39 | 52.7±0.45 | 46.2±3.63 | 54.7±0.44 | 52.3±2.26 | 53.1±0.66 |
| k=4 Topic-XICL(ours) | **63.8±0.42** | **53.4±0.26** | **55.7±0.04** | 55.5±0.22 | **56.2±0.22** | 56.3±0.44 | 56.2±0.50 | **54.4±0.43** | **56.0±0.30** | 48.1±0.44 | **52.3±0.31** | **53.3±0.20** | **50.4±0.46** | **56.0±0.47** | **54.6±0.13** | **54.8±0.13** |

Table 7: Accuracy of XCOPA in 12 languages based on BLOOM-7b1, Qwen1.5-7B and Llama-3.1-8B models.

| xcopa | | en | et | ht | id | it | qu | sw | ta | th | tr | vi | zh | AVG |
|---|---|---|---|---|---|---|---|---|---|---|---|---|---|---|
| *Qwen1.5-7B* | | 86.8 | 54.8 | 47.6 | 70.4 | 77.0 | 51.2 | 53.8 | 2.0 | 59.2 | 61.4 | 68.2 | 62.8 | 57.9 |
| k=2 | ICL_random | 89.2±0.59 | 55.0±1.61 | 49.9±0.25 | 69.2±0.59 | 70.8±2.63 | 51.6±0.57 | 51.5±1.32 | 27.7±7.74 | 59.2±2.29 | 58.0±0.59 | 66.7±3.58 | 69.1±5.75 | 59.8±0.58 |
| | ICL_sem | **91.4** | 53.6 | 52.2 | 64.4 | 70.8 | 53.2 | 52.6 | **48.4** | 62.4 | 54.2 | 63.0 | 71.6 | 61.5 |
| | ICL_cluster | 88.6±2.04 | 54.7±1.46 | 48.6±1.42 | 66.1±2.53 | 71.3±1.39 | 51.3±0.68 | 52.4±0.99 | 42.1±1.89 | 61.2±3.77 | 57.3±1.75 | 70.7±2.23 | 68.6±3.68 | 61.1±0.64 |
| | Topic-XICL(ours) | **91.4±0.16** | **58.2±0.09** | **53.4±0.47** | **70.0±0.09** | **73.8±0.28** | **54.4±0.50** | **53.6±0.50** | 47.6±0.90 | **64.8±0.47** | **58.4±0.57** | **72.0±0.81** | **77.6±2.36** | **64.6±2.36** |
| k=3 | ICL_random | 91.3±0.50 | 56.3±1.82 | 48.7±1.11 | 71.3±1.33 | 75.4±3.69 | 54.0±1.56 | 52.1±1.82 | 47.0±5.40 | 63.9±0.50 | 59.9±0.66 | 70.5±3.17 | 77.1±2.65 | 63.9±1.01 |
| | ICL_sem | 90.6 | 54.0 | **52.6** | 70.4 | 71.6 | 52.4 | 51.0 | 46.2 | 64.2 | 57.8 | 70.8 | 75.2 | 63.1 |
| | ICL_cluster | 89.9±1.09 | 54.4±2.72 | 49.7±1.09 | 69.7±1.72 | 75.8±1.02 | 51.5±1.33 | 51.5±2.61 | 50.0±0.75 | 64.5±0.62 | 57.1±0.94 | 72.4±1.13 | 74.2±2.54 | 63.4±0.70 |
| | Topic-XICL(ours) | **91.6±0.09** | **57.8±0.33** | 52.6±0.71 | **71.4±0.25** | **76.4±0.41** | **56.2±1.09** | **57.2±1.70** | **51.2±0.34** | **67.6±0.98** | **62.8±0.82** | **76.2±0.66** | **82.0±0.96** | **66.9±0.96** |
| k=4 | ICL_random | 91.1±1.05 | 55.4±1.14 | 49.5±1.06 | 69.8±0.59 | 76.5±4.30 | 52.1±1.23 | 51.7±1.27 | 50.7±1.52 | 64.8±3.02 | 59.9±1.05 | 70.7±3.62 | 80.1±0.82 | 64.3±0.68 |
| | ICL_sem | 92.0 | 51.8 | 45.2 | **71.4** | 73.6 | 53.8 | 50.6 | 50.4 | 65.2 | 56.6 | 72.4 | 75.2 | 63.2 |
| | ICL_cluster | 90.9±1.59 | 53.8±1.77 | 50.0±1.23 | 69.6±0.59 | 77.7±1.75 | 50.1±1.31 | 51.9±2.37 | 50.9±0.52 | 65.2±1.66 | 58.4±2.29 | 71.7±1.48 | 78.9±2.07 | 64.1±0.81 |
| | Topic-XICL(ours) | **92.2±0.09** | **59.2±1.06** | **51.6±0.33** | 70.6±0.25 | **77.8±0.38** | **55.4±1.55** | **54.8±0.75** | **52.2±0.34** | **69.4±1.09** | **62.6±0.59** | **75.6±0.50** | **83.2±0.09** | **67.1±0.09** |
| *BLOOM-7b1* | | 45.8 | 49.8 | 49.6 | 49.8 | 49.4 | 50.2 | 49.6 | 49.4 | 50.4 | 50 | 51 | 50.4 | 49.6 |
| k=2 | ICL_random | 56.4±0.93 | 49.5±0.34 | 49.6±1.31 | 51.6±0.9 | 50.1±0.33 | 50.9±1.33 | 49.6±1.75 | 51.6±0.75 | 52.8±0 | 53.2±0.68 | 50.9±0.82 | 49.6±0.43 | 51.3±0.4 |
| | ICL_sem | 55.2 | 49 | 52.4 | 52.4 | 47 | **52.6** | 51.2 | 49.2 | 54.2 | 53 | 49.2 | 49 | 50.8 |
| | ICL_cluster | 57.4±0.65 | 49.5±0.82 | 50.5±0.9 | 52.3±0.57 | 49.8±0.29 | 49.9±0.84 | 50.5±0.29 | 53.1±0.82 | 52.6±1.02 | 53.0±1.72 | 51.0±1.72 | 50.5±0.62 | 51.7±0.09 |
| | Topic-XICL(ours) | **58.0±0** | **53.8±0.82** | 50.4±0.66 | **53.6±1.36** | **50.8±0.9** | 52.4±0.19 | **55.0±1.32** | **53.6±1.43** | **53.8±0.93** | **54.0±1.71** | **56.8±1** | **54.8±0.62** | **53.9±0.13** |
| k=3 | ICL_random | 56.6±0.34 | 51.1±0.49 | 49.4±0.93 | 51.9±0.81 | 50.3±0.47 | 51.9±1.52 | 50.4±0.52 | 50.5±0.78 | 52.4±0.98 | 53.3±0.34 | 49.5±0.19 | 49.9±0.62 | 51.4±0.21 |
| | ICL_sem | 56 | 50 | 51 | 52.2 | 46.8 | 48.2 | 47.8 | 51.4 | 53.6 | 53.8 | 46.6 | 47.6 | 50.4 |
| | ICL_cluster | 58.2±0.29 | 48.6±0.43 | 49.7±0.52 | 52.6±0.16 | 48.2±0.47 | 49.4±1.16 | 50.3±0.68 | 48.9±0.84 | 52.9±0.96 | 53.5±0.75 | 50.7±0.57 | 48.5±0.38 | 51.0±0.26 |
| | Topic-XICL(ours) | **58.6±0.66** | **50.2±0.9** | **53.2±0.34** | **53.2±0.9** | **51.4±0.56** | **52.0±0.84** | **52.0±0.66** | **54.2±1.73** | **56.4±1.57** | **54.2±1** | **61.6±1.14** | **57.0±0.57** | **54.5±0.09** |
| k=4 | ICL_random | 57.3±0.34 | 49.6±0.68 | 49.1±0.85 | 51.7±0.9 | 49.9±0.41 | 50.1±0.57 | 50.0±0.65 | 52.1±0.73 | 52.3±1.18 | 53.7±0.1 | 49.8±0.66 | 49.9±0.65 | 51.3±0.29 |
| | ICL_sem | 56.8 | 49 | 51.6 | 51.6 | **51** | 48.8 | 49.2 | 53.2 | **54.8** | 54.6 | 50 | 47.4 | 51.5 |
| | ICL_cluster | 59.3±0.19 | 49.7±0.33 | 50.3±0.47 | 52.1±0.78 | 49.9±1.23 | 49.9±1.14 | 50.3±0.19 | 52.9±0.9 | 53.1±0.1 | 54.4±0.71 | 49.9±0.16 | 50.6±0.66 | 51.9±0.17 |
| | Topic-XICL(ours) | **59.4±0.75** | **54.0±0.87** | 50.4±1 | **53.0±0.34** | 50.8±1.39 | **51.0±1.14** | **55.6±0.49** | **53.2±0.34** | 53.6±1.96 | **55.2±0.85** | **58.6±1.7** | **58.0±0.41** | **54.4±0.16** |
| *Llama-3.1-8B* | | 86.8 | 60.2 | 52.2 | 78.4 | 77.6 | 50.2 | 53.8 | 64.2 | 65.0 | 68.0 | 75.0 | 79.2 | 67.6 |
| k=2 | ICL_random | 94.8±0.28 | 64.2±2.01 | 54.9±0.90 | 83.5±1.54 | 86.9±0.41 | 50.9±0.84 | 59.7±2.90 | 66.6±2.44 | 74.2±4.55 | 71.3±2.00 | 81.1±1.32 | 86.7±1.00 | 72.9±1.50 |
| | ICL_sem | 95.0 | 62.4 | 54.2 | 80.4 | 85.4 | 49.8 | 61.0 | 67.0 | 70.2 | 69.8 | 79.8 | 83.4 | 71.5 |
| | ICL_cluster | 94.8±0.59 | 63.3±1.52 | 55.9±0.41 | 82.4±1.66 | 86.3±0.41 | 50.5±0.57 | 61.6±0.59 | 67.3±0.50 | 71.5±0.90 | 73.5±0.52 | 81.1±0.38 | 85.9±0.74 | 72.9±0.25 |
| | Topic-XICL(ours) | **95.6±0.19** | **66.0±0.77** | **57.0±0.25** | **84.6±0.68** | **88.4±0.25** | **52.6±0.57** | **62.2±0.25** | **69.6±0.81** | **76.0±0.85** | **74.0±0.16** | **82.4±0.25** | **88.0±0.52** | **74.7±0.52** |
| k=3 | ICL_random | 95.4±0.28 | 63.7±2.32 | 54.9±1.48 | 82.6±1.31 | 87.8±1.34 | 50.9±1.05 | 60.1±3.13 | 66.5±1.52 | 74.9±4.20 | 70.9±2.71 | 82.4±1.70 | 87.3±0.90 | 73.1±1.46 |
| | ICL_sem | 94.8 | 62.8 | 54.0 | 81.8 | 86.6 | 50.2 | 58.8 | 66.2 | 74.6 | 71.2 | 81.6 | 83.4 | 72.2 |
| | ICL_cluster | 95.4±0.43 | 63.3±1.80 | 56.3±0.34 | 83.1±1.39 | 86.9±0.68 | 51.5±0.50 | 60.0±0.16 | 68.2±0.82 | 74.5±0.74 | 72.7±1.15 | 81.9±0.81 | 86.9±1.16 | 73.4±0.25 |
| | Topic-XICL(ours) | **96.0±0.09** | **65.8±0.41** | **56.8±0.16** | **84.8±0.43** | **88.6±0.25** | **52.2±0.34** | **62.4±0.50** | **70.0±0.43** | **77.6±0.71** | **74.2±0.81** | **83.4±0.38** | **88.8±0.41** | **75.0±0.41** |
| k=4 | ICL_random | 95.3±0.68 | 63.6±1.23 | 54.7±0.98 | 83.7±0.98 | 88.1±1.36 | 50.4±1.02 | 60.4±1.88 | 66.5±0.98 | 75.1±2.16 | 72.1±1.68 | 83.0±1.30 | 87.2±0.98 | 73.3±1.00 |
| | ICL_sem | 95.4 | 62.2 | 53.4 | 82.8 | 86.0 | 51.2 | 61.0 | 66.8 | 73.6 | 69.6 | 82.4 | 82.8 | 72.3 |
| | ICL_cluster | 95.7±0.41 | 63.7±1.86 | 55.8±0.28 | 83.5±1.20 | 87.2±0.98 | 51.5±1.60 | 60.7±0.38 | 66.9±0.09 | 73.9±0.57 | 72.8±1.45 | 81.0±1.18 | 87.9±1.09 | 73.4±0.48 |
| | Topic-XICL(ours) | **96.2±0.19** | **66.2±0.28** | **56.2±0.09** | **85.4±0.62** | **89.4±0.09** | **53.8±0.75** | **62.8±0.81** | **71.0±0.94** | **76.4±0.34** | **74.2±0.57** | **84.6±0.65** | **89.4±0.52** | **75.5±0.52** |

Table 8: Accuracy of XNLI in 15 languages based on BLOOM-7b1, Qwen1.5-7B and Llama-3.1-8B models.

