# OpenReview forum: "Topic-XICL: Demonstration Selection with Topic Inference for Cross-lingual In-context Learning"
_ICLR.cc/2025/Conference — Submitted to ICLR 2025_

### Official Review · Reviewer_yieg · 2024-10-31

**Soundness:** 2
**Presentation:** 3
**Contribution:** 3
**Rating:** 5
**Confidence:** 4

**Summary:**

This work utilizes the K-means algorithm for clustering to form topics and fine-tunes a LLM by introducing new tokens, creating cross-language semantic representations of topics. During the inference phase, the topic is determined based on the input, and relevant use cases that match the topic and share similar meanings are retrieved as examples to improve in-context performance.

**Strengths:**

1. The training on cross-language topics facilitates knowledge transfer across languages, achieving significant performance improvements in multiple multilingual tasks compared to traditional methods, particularly in low-resource scenarios.
2. It validates the importance of topic similarity in constructing demonstrations, suggesting that this is more effective than relying solely on semantics, which is an interesting finding.
3. The experiments are thorough, with validation of the method across multiple languages and tasks.

**Weaknesses:**

1. The most critical part is using the LLM to fine-tune the token representations across languages for topic modeling. However, the issue of data ratio for topics in different languages was not introduced. How can we ensure that the training achieves the best results across multiple languages?
2. The determination of topic categories relies on clustering, how is the optimal number of topics determined?
3. The clustering process for forming training data may result in a significant portion of the data not being clustered, leading to considerable disparities in topic distribution. How well the model performs on less common topics has not been analyzed in depth.
4. There is a lack of recent comparisons with related models for demonstration selection, such as TopK + MDL, TopK + ConE, etc.

**Questions:**

see above

---

### Official Review · Reviewer_QHLi · 2024-11-02

**Soundness:** 3
**Presentation:** 3
**Contribution:** 3
**Rating:** 6
**Confidence:** 3

**Summary:**

The paper introduces Topic-XICL, a novel demonstration selection algorithm designed to enhance cross-lingual in-context learning (XICL) by leveraging latent topic inference. Traditional methods for demonstration selection often rely solely on semantic similarity or task-specific retrievers, which may not capture the nuanced diversity required for complex multilingual tasks. Topic-XICL addresses this by integrating richer diversity information through latent topic variables, mined using large language models (LLMs). The proposed framework clusters source language data into latent topics, learns topic embeddings, and selects the most representative demonstrations for each target input based on topic inference. The method is validated across three datasets—XNLI, XCOPA, and TyDiQA-GoldP—and three LLMs—BLOOM, Qwen1.5, and Llama3.1—showing consistent improvements, particularly in low-resource and unseen languages.

**Strengths:**

1. The integration of latent topic inference with Bayesian theory for demonstration selection in XICL is a novel approach that extends existing methodologies beyond simple semantic similarity.
2. Comprehensive experiments across multiple datasets and models provide strong evidence of the method's effectiveness.
3. The method is versatile, applicable to a wide range of tasks beyond classification, and demonstrates significant improvements in low-resource and unseen languages.

**Weaknesses:**

1. The explanation of the Bayesian framework and latent topic modeling could be more accessible. Simplifying the theoretical sections or providing additional intuitive explanations would benefit readers.
2. While empirical guidelines are provided, a deeper analysis of how hyperparameters like the number of topics (n) and tokens (c) affect performance across different tasks could offer more insights.
3. Including additional baseline methods for demonstration selection could provide a more comprehensive evaluation of Topic-XICL's performance.

**Questions:**

How sensitive is Topic-XICL to the choice of the number of topics (n) and the length of topic tokens (c)?

---

### Official Review · Reviewer_QG6z · 2024-11-03

**Soundness:** 3
**Presentation:** 1
**Contribution:** 2
**Rating:** 3
**Confidence:** 3

**Summary:**

The paper presents a method for selecting demonstrations in cross-lingual in-context learning. In the first stage, topic-specific features are learned for a high-resource language by adding topical tokens. Then, the *k*-means algorithm clusters candidates into *n* topics based on these tokens, and top-*k* representative demonstrations are selected for each topic. In the next stage, for each target input, a similarity score is calculated for these representative demonstrations. Majority voting is then applied to the top 10 demonstrations to identify the most relevant topic for the target input. Finally, during inference, only the representative demonstrations from the selected topic are used. Results indicate that this method improves performance compared to baseline approaches that rely on similarity and diversity alone.

**Strengths:**

1. The results demonstrate improvements over widely used baselines, such as those that select demonstrations based on similarity scores and diversity.
2. The experiments include a wide range of languages, making the study comprehensive.

**Weaknesses:**

1. The paper lacks sufficient novelty to be considered a standalone contribution. Most of the proposed idea replicates the work of Wang et al. (2023), with a minor modification involving topic selection based on majority voting. In fact, the novelty of this adjustment is relatively marginal compared to the original idea.

2. The presentation is difficult to follow, as the paper feels somewhat disjointed in several areas, especially in how the proposed method is presented: (1) In lines 219-220, the paper mentions that topical tokens are *appended/concatenated* to the input, but both the example and Figure 2 indicate that the tokens are actually *prepended*; (2) The title of Figure 2 does not explain topical tokens, leaving their purpose and role unclear; (3) The abstract makes no mention of a Bayesian inference perspective, yet this concept is introduced abruptly in the main text; and (4) In lines 75-89, where the method is explained, there is no mention of using majority voting to select the relevant topic for the target input. This makes the method appear different from what is described in the abstract and method sections.

3. There are more typographical and grammatical issues than usual. For instance, "updata embedding" appears in Figure 2, there is a missing period in the title of Figure 3, and inconsistent capitalization in terms like "K-means" vs. "k-means," among other errors.
---
Reference:

[1] Large Language Models Are Latent Variable Models: Explaining and Finding Good Demonstrations for In-Context Learning (Wang et al., NeurIPS 2023)

**Questions:**

As mentioned in the paper, $a$ and $c$ are hyperparameters that need to be determined. My question is whether the authors chose the values used in the paper for these hyperparameters randomly or if those values represent the best results they found in their experiments. If the values were selected randomly, how does performance change based on different values for these hyperparameters?

---

### Official Review · Reviewer_wy6m · 2024-11-04

**Soundness:** 3
**Presentation:** 2
**Contribution:** 2
**Rating:** 3
**Confidence:** 4

**Summary:**

This paper addresses the demonstration selection for cross-lingual ICL. Specifically, Topic-XICL is proposed to train a latent topic model on demonstration candidates from rich-resource language and select top k representative examples corresponding to the target input's topic. Experiments are conducted on three LLMs on three multilingual datasets.

**Strengths:**

- Empirical results show that Topic-XICL surpasses baseline methods on three multilingual datasets, demonstrating the effectiveness of the proposed method.

**Weaknesses:**

- The major concern of this paper is that it appears to be an incremental work. The paper mainly extends the demonstration selection method in [1] to a different setting (XICL), which may not contribute significantly to the field.
- There is a lack of comparison with SOTA works in the main experiment. Although several SOTA works are mentioned in Related Work, they are not included in the empirical evaluation. This makes it difficult to evaluate the effectiveness of the proposed method relative to existing methods.
- Additional computational costs are introduced in training the topic model and demonstration selection, however, there is a lack of analysis and quantitative comparison of the efficiency of the proposed method.
- The paper lacks experiments on the impact of varying the number of examples (larger k values) and testing LLMs of different sizes, which could provide insights into the scalability of the method.

---
[1] "Large language models are implicitly topic models: Explaining and finding good demonstrations for in-context learning." arXiv preprint.

**Questions:**

n/a

---

### Meta-Review · Area_Chair_MWDK · 2024-12-18

**Metareview:**

This paper introduces Topic-XICL, a method for improving demonstration selection in cross-lingual in-context learning through latent topic modeling. The authors propose training a topic model on rich-resource language data and applying it for demonstration selection across various LLMs. The work shows empirical improvements over baseline methods on three multilingual tasks (XNLI, XCOPA, and TyDiQA-GoldP) using three different LLMs, with particular benefits for low-resource and unseen languages.

While the paper demonstrates consistent performance improvements and includes comprehensive experiments across multiple datasets and models, several significant limitations prevent its acceptance. The work appears largely incremental, primarily extending an existing demonstration selection method to the XICL setting without substantial theoretical innovation. Furthermore, the paper lacks comparison with state-of-the-art methods in the main experimental evaluation and contains notable presentation issues, including inconsistencies in the method description and missing analyses of computational costs and hyperparameter sensitivity.

Given these limitations and the lack of author engagement during the discussion period (no rebuttal was provided), this submission does not meet the bar for acceptance at ICLR 2025. While the topic is relevant and the empirical results show promise, the paper requires substantial revision to address its current shortcomings.

**Additional Comments On Reviewer Discussion:**

The reviewers raised several important concerns that went unaddressed due to the absence of author response during the rebuttal period. Reviewer QG6z and wy6m highlighted the incremental nature of the work, particularly its similarity to Wang et al. (2023). Reviewer QHLi requested clarification on the sensitivity of Topic-XICL to hyperparameter choices, while Reviewer yieg raised concerns about topic distribution and data ratios across languages. Additional concerns included the lack of SOTA comparisons and presentation issues in the methodology section. The authors' decision not to participate in the discussion period left these critical points unaddressed, which significantly influenced the final rejection decision.

---

### Decision · Program_Chairs · 2025-01-22

Reject